# Automatic Identification of Failure in Hip Replacement: An Artificial Intelligence Approach

**DOI:** 10.3390/bioengineering9070288

**Published:** 2022-06-29

**Authors:** Mattia Loppini, Francesco Manlio Gambaro, Katia Chiappetta, Guido Grappiolo, Anna Maria Bianchi, Valentina D. A. Corino

**Affiliations:** 1Department of Biomedical Sciences, Humanitas University, Via Rita Levi Montalcini 4, 20090 Pieve Emanuele, MI, Italy; francesco.gambaro23@gmail.com; 2IRCCS Humanitas Research Hospital, Via Alessandro Manzoni 56, 20089 Rozzano, MI, Italy; katia.chiappetta@humanitas.it (K.C.); guido.grappiolo@humanitas.it (G.G.); 3Fondazione Livio Sciutto Ricerca Biomedica in Ortopedia—ONLUS, Via A. Magliotto 2, 17100 Savona, SV, Italy; 4Department of Electronics, Information and Bioengineering, Politecnico di Milano, Via Golgi 39, 20131 Milan, MI, Italy; annamaria.bianchi@polimi.it (A.M.B.); valentina.corino@polimi.it (V.D.A.C.); 5Cardio Tech-Lab, Centro Cardiologico Monzino IRCCS, Via Carlo Parea 4, 20138 Milan, MI, Italy

**Keywords:** hip prothesis, machine learning, orthopedics, bioengineering

## Abstract

Background: Total hip arthroplasty (THA) follow-up is conventionally conducted with serial X-ray imaging in order to ensure the early identification of implant failure. The purpose of this study is to develop an automated radiographic failure detection system. Methods: 630 patients with THA were included in the study, two thirds of which needed total or partial revision for prosthetic loosening. The analysis is based on one antero-posterior and one lateral radiographic view obtained from each patient during routine post-surgery follow-up. After pre-processing for proper standardization, images were analyzed through a convolutional neural network (the DenseNet169 network), aiming to predict prosthesis failure. The entire dataset was divided in three subsets: training, validation, and test. These contained transfer learning and fine-tuning algorithms, based on the training dataset, and were implemented to adapt the DenseNet169 network to the specific data and clinical problem. Results: After the training procedures, in the test set, the classification accuracy was 0.97, the sensitivity 0.97, the specificity 0.97, and the ROC AUC was 0.99. Only five images were incorrectly classified. Seventy-four images were classified as failed, and eighty as non-failed with a probability >0.999. Conclusion: The proposed deep learning procedure can detect the loosening of the hip prosthesis with a very high degree of precision.

## 1. Introduction

This work is an attempt to use an artificial intelligence approach to solve an identified clinical need in the orthopedic field: the necessity for early recognition of total hip arthroplasty (THA) failure. THA is extremely widespread worldwide, as a highly effective treatment for several hip diseases in both young and elderly people. In Italy, the number of primary hip replacements increased from 66,560 in 2001 to 97,263 in 2016 with an average increase of 3.1% per year [1]. Despite being more common in elderly patients, approximately 25% of patients undergoing joint replacement are under 65 years old. A further increase in primary prosthetic implants placement is foreseen within 2030, due to progressive population aging and the growing number of procedures in younger patients. For the same reasons, a significant increase of implant revisions is to be expected [2,3]. Aseptic loosening, bearing surface wear, and osteolysis have been found to be the most common causes of THA failure; thus, a properly conducted radiographic follow-up aims to ensure earlier identification of complications and failures, which are likely to be manageable with more conservative revisions, leading to favorable functional outcomes. However, the detection of loosening still remains a challenge, and the final diagnosis is often confirmed at the time of revision surgery. The use of artificial intelligence (AI), particularly the deep learning (DL) approach, to perform the automatic evaluation of X-ray imaging for monitoring patients with hip arthroplasties could enhance the diagnostic accuracy for THA failure. Automatic classification algorithms are tools thought to deploy the informative content within a large amount of data. DL is a research field in automatic learning based on hierarchical levels and different concept representations. The resulting complex processing structures are grouped under the general name of convolutional neural networks (CNN) employed for object detection and image classification [4]. DL methods have already been applied to plain film radiographs with a high degree of success in different orthopedic applications, such as the identification of wrist, elbow, ankle, and hip fractures [5], and as a classification of knee osteoarthritis [6]. The aim of this study was to develop the DL algorithm and the related preprocessing pipeline to automatically detect hip prosthetic loosening from a conventional plain radiograph and to employ the algorithm on a cohort of radiographs to analyze its performance in terms of sensitivity, specificity, and accuracy. 

## 2. Materials and Methods

### 2.1. Sample

The current work has been carried according to the Strengthening the Reporting of Observational Studies in Epidemiology (STROBE) checklist [7]. Patients included in this study were retrospectively collected from the digital medical records at a tertiary academic medical center between 2009 and 2019. Patients with hip prothesis, cemented or uncemented, who underwent total or partial revision due to implant failure in the considered period were included (failed group). Implant failure was defined by the presence of either stem loosening, acetabular cup loosening, malpositioning of the implant, polyethylene wearing, or periprosthetic infection. At the radiographic assessment, stem loosening was defined as a progressive axial radiolucency greater than 3 mm, or a varus/valgus deviation from the femoral shaft axis greater than 3° [8]. Instead, the loosening of the acetabular cup was defined by a change greater than 2 mm in the horizontal and/or vertical position with an adjacent radiolucent zone, or a radiolucent zone greater than 3 mm [5]. The malposition of the implant was defined by prosthetic, bony, or soft tissue impingement of the implant [9]. In patients with fixed metal on metal implants, revision surgery was performed for a large thick-walled pseudotumor at MRI, or for extremely high metal ion levels (>10–20 ppb) in the serum or whole blood [10]. Polyethylene wear is defined by the eccentric position of the femoral head with respect to the acetabular cup in the AP and/or lateral view. A control group (non-failed group) was also included by randomly collecting patients who underwent cemented or uncemented primary total hip replacement in the same period with a rate of 1:2. To be finally included in the study, a minimum of one antero-posterior (AP) and one lateral (LAT) radiographic view of the implant needed to be available before revision surgery for the failed group and during follow-up time for the non-failed group. When patients of the non-failed group had THA in both hips, all the implants were used for the analysis. The radiographs were collected in their original resolution. The study was approved by the Institutional Ethical Committee, and all patients gave their written informed consent.

### 2.2. Model Development

#### 2.2.1. Preprocessing

All images (in DICOM format) were processed in order to have the same size and the same pixel range (values between 0 and 1). Frontal images were first split vertically into two parts, so that each one included only one limb (Figure 1a). Subsequently, the mist-like effect was reduced through increased brightness, implemented through gamma power transformation (Figure 1b) [1]. A sigmoidal function was adopted to improve the contrast (Figure 1c). Finally, the contrast-limited adaptive histogram equalization method allowed for enhancing the contrast [11]. Unlike conventional histogram equalization, it operates on small data regions (tile) rather than on the entire image. The neighboring tiles were then combined using bilinear interpolation (Figure 1d) [11]. Finally, the image was resized to a standard input dimension (224 × 224) and was standardized by z-score. A data augmentation technique was applied to increase the number of images in the training set by making a number of non-exact copies, or transformations, of each image. This step provided the network with more training examples by incorporating the salient features in multiple orientations. Transformations included horizontal flip through 180° rotation, rotation (range 0–30°), and zoom (range −20–20%). This resulted in an overall increase in the training set approximately by a factor of 3. Performing data augmentation allowed for making the resulting model more robust to non-relevant sources of variability, including suboptimal positioning of patients within the radiograph and suboptimal exposure settings.

#### 2.2.2. CNN Training

Models were developed using Tensorflow 1.15, an opensource software library for machine intelligence. A CNN is a DL algorithm that can take an image as the input, assign importance (learnable weights and biases) to different features (parameters/objects) in the image, and then differentiate one from the other. A CNN is able to successfully capture the spatial dependencies in an image, through the application of relevant filters in parallel, and to classify them based on this. A CNN is composed of various types of layers (or building blocks): each layer is used to collect, summarize, and/or transform data before passing them to the following layer. The DenseNet169 network [12], trained for the IMageNet Large Visual Recognition Challenge [13] and based on the analysis of non-radiological images, was used in this study. DenseNet architecture shows layers that are all directly connected with each other. This main advantage grants features reuse from different levels, thus improving computational, memory efficiency, and focus on the problem of interest. DenseNet was computationally adapted to produce models for prosthesis failure recognition by using transfer learning and fine tuning algorithms [14]. Transfer learning consists of reusing a model developed for a task as the starting point for a model on a second task, i.e., some layers are frozen from the original task, while others are trained for the new one. It is a common and effective strategy to use transfer learning from a network pretrained on an extremely large dataset and to then reuse it for a different task of interest, usually containing less images. The underlying assumption of transfer learning is that generic features learned on a large enough dataset can be shared among similar or even different datasets, thus improving generalization. 

Regarding output, the network provides the probability for the image to belong to the failed or to non-failed group. A stratified data split procedure was employed: 10% of samples were used for the testing phase, and the remaining samples were furtherly split in 70% and 30% for training and validation, respectively. The stratified split was random, though maintaining the proportion of failed and non-failed images in all of the three groups. Finally, for the test set, the Gradient weighted Class Activation Mapping (Grad-CAM) tool [15] was used to highlight which parts of the image mostly contributed in the classification, thus putting into evidence the image regions in which the algorithm bases its classification choice. To assess the network performance, the area under the receiver operator characteristic (ROC) curve (AUC) was calculated, as well as the accuracy, sensitivity, and specificity.

## 3. Results

Six hundred thirty patients (average age 72 years—range 26 to 88—40% males) were retrospectively analyzed. Of these patients, 420 had prosthesis subsequently requiring a revision (failed group) and 210 had prosthesis not requiring further surgery (non-failed group). In the failed group, 224 patients underwent acetabular revision, 138 underwent acetabular and stem revision, 51 underwent stem revision, 2 underwent Girldestone operation, 2 underwent removal of the prothesis due to periprothesic infection, 1 underwent femoral head revision, 1 underwent change the polyethylene liner, and 1 required a trochanteric screw. Forty six patients (10.9%) of the failed group had a cemented implant. Ninety-six patients had a non-failed bilateral implant (thus images from both hips were included in the analysis) and some patients had more than two X-ray examination, thus the final dataset included 1853 images grouped in 922 failed and 931 non-failed prostheses. The progression of the performance of our model in the analyzed data sets demonstrated that the accuracy decreased slightly from the training to the validation and test sets, in all cases being higher than 0.96 (Figure 2 and Figure 3). In the test set, the AUC was 0.99 and the ROC curve was almost equal to the ideal curve (Figure 4). The classification was correct in 97% of cases, with only five images incorrectly classified. Seventy-four images were classified as failed, and 80 images were classified as non-failed with a probability > 0.999 (Figure 5). For the remaining images of the test set, the probability is shown in the Figure 3. Among the five images incorrectly classified, two were classified as non-failed with a probability of 0.82 and 0.85, respectively; in particular, in these two cases there was an acetabular cup failure that was rated as normal. On the other hand, the three images that were incorrectly classified as failed had a probability of 0.77, 0.98, and >0.999 (Table 1); in particular, the system reported in hip prothesis cerclage but there was no pathological finding. Therefore, the algorithm showed a sensitivity of 96.7% and a specificity of 96.7%. Figure 6 shows an example of the Grad-CAM for failed and non-failed prosthesis. It can be observed that for the non-failed image, no regions close to the prosthesis were lit up; on the contrary, in the failed images, pathological traits (i.e., the possible cause of failure) are highlighted. These results were confirmed on the whole test set, with the results of the Grad-CAM being reviewed by an expert orthopedic surgeon. The images of the failed prosthesis in the test set were 94: the algorithm identified a wrong area in 5 images, in 12 it identified only partially the pathological areas, and in 77 images it correctly identified the pathological traits.

## 4. Discussion

The main finding of the present study was that a DL algorithm applied to plain radiographs is able to detect the loosening of the hip prosthesis with a very high degree of precision (>0.97%). Therefore, the DL algorithms could be applied in the follow-up of patients with hip replacement as a tool for the detection of implant failure. The follow-up assessment of hip and knee replacement is currently done with conventional X-rays, and it mainly aims to detect component malalignment, subsidence, prosthesis loosening and polyethylene wear. The early detection of these complications on the basis of two-dimensional (2D) images (X-rays) can be highly challenging for clinicians. In order to overcome this limitation, an attempt has been made to develop computer-based image analysis methods. The roentgen stereophotogrammetric analysis (RSA) is a method allowing for a reliable measurement of the micromotion, both for the prosthesis components and for the prosthesis itself compared with the bone [16]. Besides micromotion, it can also provide a reliable estimation of polyethylene wear. In the marker-based RSA, the analysis is performed through prearranged tantalum markers embedded into bones or attached to prostheses [17]. The major drawbacks of this technique include the need for a specific setup, for a surgical preparation, costs, and ethical issues related with the insertion of tantalum markers. In addition, the prosthetic components can potentially shade the markers. In the model-based RSA, the analysis is performed through the acquisition of prosthetic models and two X-ray images (anteroposterior (AP) and lateral views) of the prosthesis [18]. Although this technique avoids the use of tantalum markers, it demonstrated a worse accuracy compared with the marker-based RSA. The Einzel-Bild-Roentgen-Analyse (EBRA) method was developed to assess the migration of the acetabular cup and femoral head by using standard AP views of the pelvis [19]. EBRA requires X-rays to be grouped in different comparable sets, to be used to evaluate micro-movements. Images are defined as comparable if there is an overlap of different landmark pelvic references. Although it allows for good spatial reconstruction, EBRA use is often limited by the lack of comparable X-rays. Because the final diagnosis of prosthetic loosening is still a challenge, particularly in the early stages, interest in the use of AI-based algorithms as a diagnostic tool is increasing overtime. In a recent study, Shah et al. [20] developed a machine-learning algorithm to diagnose hip and knee prosthetic loosening by using preoperative radiographs and patient features. They demonstrated that the model developed with the combination of both types of data reported an accuracy, sensitivity, and specificity of 88.3%, 70.2%, and 95.6%, respectively, whereas the performance of the model based on radiographs alone was worse with an accuracy of 70%. The algorithm developed in this study showed a sensitivity of 96.7%, a specificity of 96.7%, and an accuracy of 96.7%. The best performance of the present algorithm could be explained by the use of images of patients with both failed and non-failed implants. We hypothesize that the use of both images allowed for better detection of the radiological signs of loosening. On the other hand, Shah et al. included only patients who underwent primary hip or knee revision arthroplasty, and used the gold-standard diagnosis of fixation for the intraoperative findings of fixed or loose implants. Moreover, in the present study twice the number of failed patients compared with the previous study were included, resulting in a greater number of images used for the development of the algorithm. Despite the good performance of the present algorithm, five images were misclassified. In one case (Figure 7a), the algorithm did not detect a stem osteolysis; a possible explanation for this mistake could be that actually the stem remained in place despite the osteolysis, while usually, together with the osteolysis, a stem loosening is noted. In another case (Figure 7b), the cerclage wiring around the prothesis was classified as pathological by the algorithm, instead the implant had not failed. In another case (Figure 7c), the algorithm did not detect the polyethylene insert wearing, indirectly represented in the X-ray film by an eccentric head in relation to the cup. A possible explanation for this is that polyethylene insert wearing as the main reason for revision was an infrequent cause for failure in our data set (0.01%); therefore, the algorithm may have not been trained enough to recognize this type of failure and, moreover, in this specific case, the degree of wearing was not as severe as the other cases present in the dataset. In another case (Figure 7d), stem loosening was not detected, and, finally, in the last case (Figure 7e), a severe cup loosening with luxation was not highlighted by the algorithm. 

Some limitations affect the present study. 

First of all, the radiographs used for algorithm development were retrospectively collected from patients undergoing partial or total hip replacement revision and patients who underwent primary THA without any clinical or radiographic signs to suspect the failure of the implant. Therefore, the algorithm has not been tested in patients who had a clinical or radiographic suspicion of loosening, but did not have surgery. Thus, the current value of the accuracy, sensitivity, and specificity could not be confirmed in that population. Further studies are needed to investigate the role of AI algorithms in patients with an uncertain preoperative diagnosis of implant loosening. Finally, no standardized criteria for the preoperative radiographic diagnosis of loosening were applied. The gold-standard of loosening was represented by revision surgery; therefore, it could be biased by the assessment of loosening performed by different surgeons. Future prospective studies should include improvement of the algorithm for the diagnosis of prosthetic failure with the association of radiological and clinical data. The increase in the burden of joint replacements, the need for centralization, and the amount of clinical and radiological assessments required make regular follow-up of prostheses difficult to be sustained in the long run. Some authors investigated the role of a “virtual clinic”, characterized by a remote review of questionnaires and radiographs from the orthopedic surgeon in order to reduce the burden of follow-up consultations [21,22]. In the future, AI algorithms could be used in the virtual clinic in order to distinguish patients who can continue a virtual follow-up from patients requiring a face-to-face visit because of pathological clinical or radiological signs. In conclusion, the proposed DL procedure based on plain radiographs can detect the loosening of the hip prosthesis with a very high degree of precision. Thus, the proposed algorithms could find application in the follow-up of patients with hip replacement as a tool for the identification of implant failure.

## Figures and Tables

**Figure 1 bioengineering-09-00288-f001:**
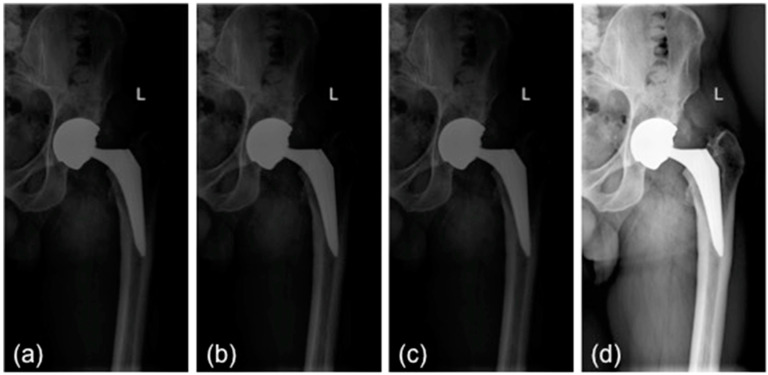
Preprocessing steps: (**a**) initial image; (**b**) mist effect reduction; (**c**) contrast enhancement; (**d**) contrast-limited adaptive histogram equalization. L: left side.

**Figure 2 bioengineering-09-00288-f002:**
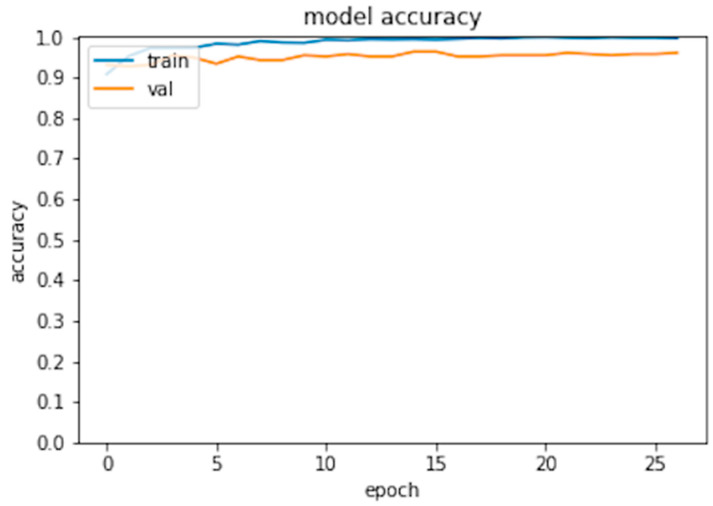
Model accuracy for the training and validation set.

**Figure 3 bioengineering-09-00288-f003:**
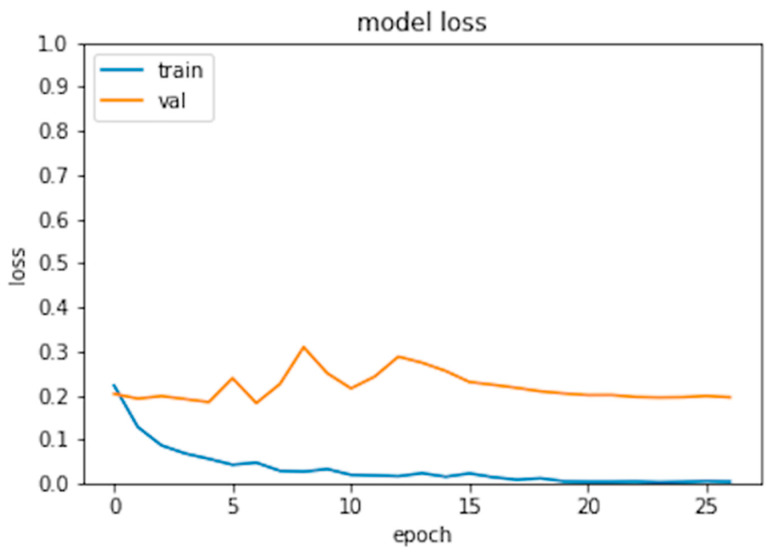
Model loss for the training and validation set.

**Figure 4 bioengineering-09-00288-f004:**
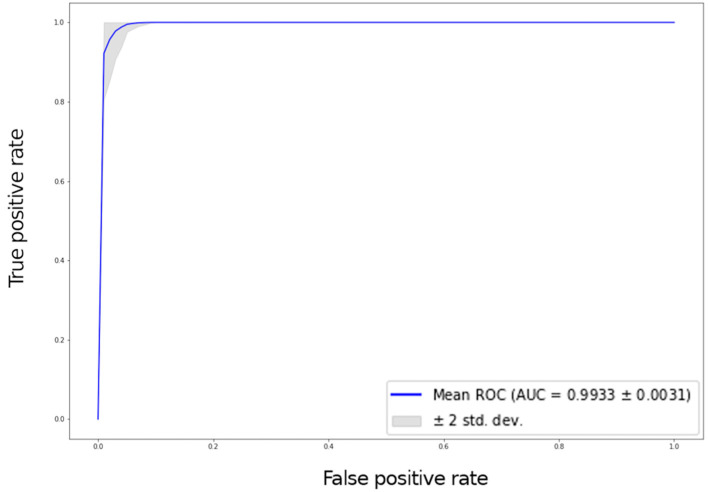
ROC curve with a 95% confidence interval (grey area) for the test set.

**Figure 5 bioengineering-09-00288-f005:**
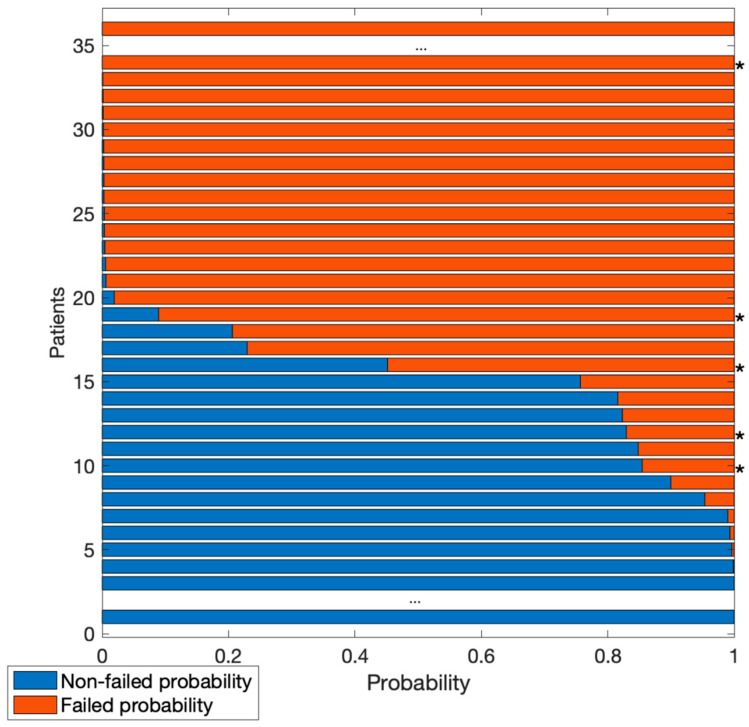
Probability of images in the test set to be classified as failed (red) or non-failed (blue). Central dots indicate many images with a probability >0.999 of belonging to one of the two classes. Asterisks (*) indicate images incorrectly classified.

**Figure 6 bioengineering-09-00288-f006:**
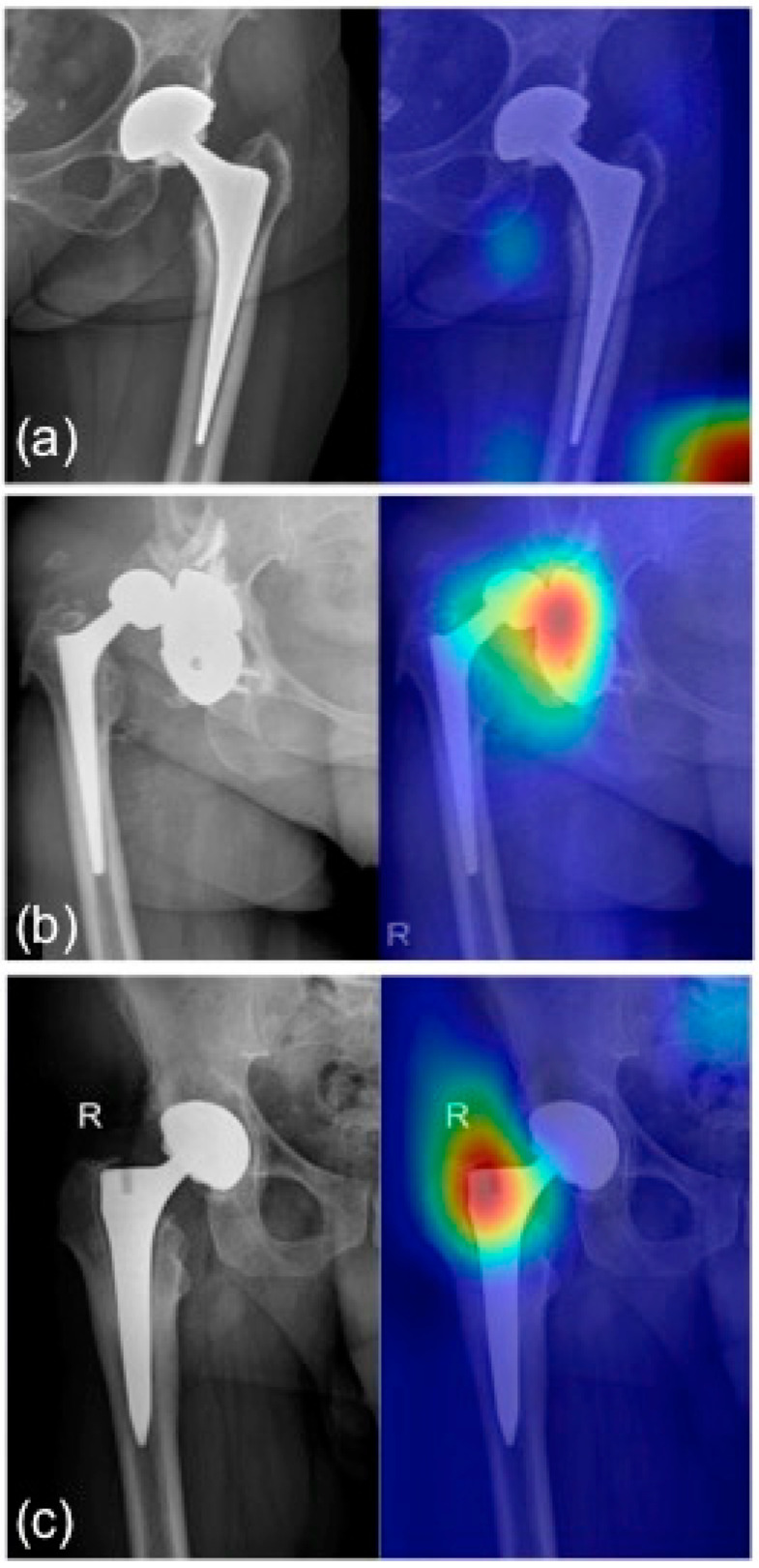
Grad-CAM for (**a**) healthy prosthesis heatmap: no regions in the image are lighted up with warm colors; (**b**) pathological acetabular heatmap: acetabular cup dislocation glows in red; (**c**) pathological stem heatmap: pathological traits are highlighted in the stem region. R: right side.

**Figure 7 bioengineering-09-00288-f007:**
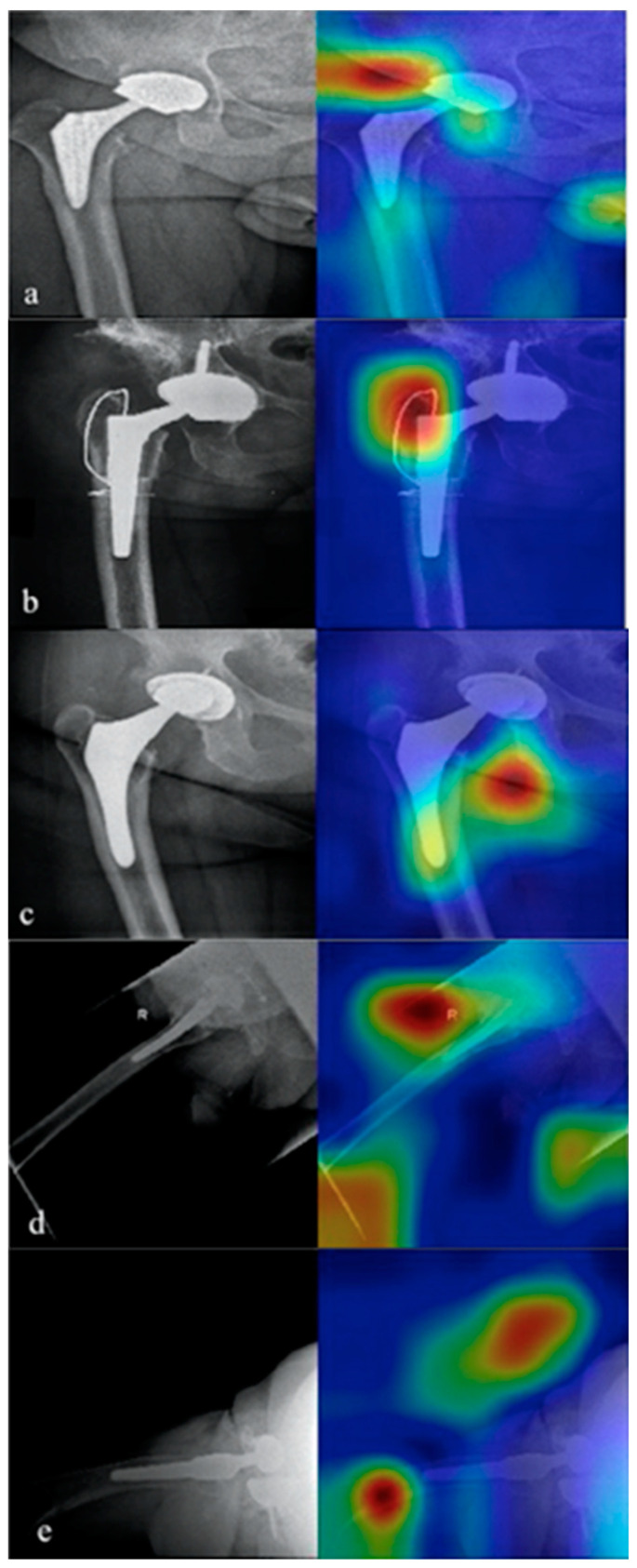
The five misclassified images. In (**a**) the algorithm did not detect a stem osteolysis. In (**b**) the cerclage wiring around the prothesis was wrongly classified as pathological by the algorithm. In (**c**) the algorithm did not detect the polyethylene insert wearing. In (**d**) the stem loosening was not detected. In (**e**) a severe cup loosening with luxation was not detected.

**Table 1 bioengineering-09-00288-t001:** Performance in training, validation, and test sets.

Set	Metrics	
Training	Accuracy	0.99
Validation	Accuracy	0.975
Test	Accuracy	0.9677
	Sensitivity	0.9677
	Specificity	0.9677
	AUC	0.9933

## Data Availability

The data repository for this study will be shortly available since we are currently working on it.

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
