# Peer review of "Automatic Identification of Failure in Hip Replacement: An Artificial Intelligence Approach"

_bioengineering, 2022, doi:10.3390/bioengineering9070288_

Round 1

Reviewer 1 Report

This paper is a case report, I think it matches with the Aim of Bioengineering  of Translational Case Reports, and its Scope of Biomedial Diagnosis. The topic is appropriated for publication in Bioengineering.

The methodology of the experiment is good, the available data for train, validate and test the CNN is really valuable. Also, the data augmentation technique was a smart option for increasing the robustness of the algorithm. For comparison, it will be great if you have results without the data augmentation and include them in the document.

In model development, the description of the preprocessing of film radiographs is clear and detailed. But it is not so with the explanation of the CNN training. The authors only mentioned that they used a well-known Convolution Neural Network that was adapted for others to achieve prosthesis failure recognition by using transfer learning and fine-tuning algorithms. It is clear the development of the CNN is not the focus of this paper, and it is only a tool for analyzing data in this Case Report. However, a more detail explanation about this CNN and the process for adapt it to radiological images is highly desirable.

Figure 4(a). Authors mentioned that it is a healthy prosthesis and no regions where lighted up, but in the bottom-right side there is a red zone that far from the prosthesis. What happened here? other similar images presented the same results. Additionally, what is the explanation for the three light blue circles around the prosthesis in the same Figure 4(a)? I can appreciate a similar light blue circle in the top-right side of Figure 4(c).

The quality of Figures 2 o 3 must be improved.

Author Response

This paper is a case report, I think it matches with the Aim of Bioengineering  of Translational Case Reports, and its Scope of Biomedial Diagnosis. The topic is appropriated for publication in Bioengineering.

The methodology of the experiment is good, the available data for train, validate and test the CNN is really valuable. Also, the data augmentation technique was a smart option for increasing the robustness of the algorithm. For comparison, it will be great if you have results without the data augmentation and include them in the document.

R: We thank the Reviewer for the appreciation of our work. Data augmentation is a very common step used when training CNN, to avoid overfitting and obtain more robust results. For this reason, we decided not to include results without data augmentation in our manuscript, as we would not have enough data to train the CNN.

In model development, the description of the preprocessing of film radiographs is clear and detailed. But it is not so with the explanation of the CNN training. The authors only mentioned that they used a well-known Convolution Neural Network that was adapted for others to achieve prosthesis failure recognition by using transfer learning and fine-tuning algorithms. It is clear the development of the CNN is not the focus of this paper, and it is only a tool for analyzing data in this Case Report. However, a more detail explanation about this CNN and the process for adapt it to radiological images is highly desirable.

R: We added information on CNN in the CNN training section, that now reads “A CNN is able to successfully capture the spatial dependencies in an image, through the application of relevant filters in parallel, and to classify based on them. A CNN is composed of various types of layers (or building blocks): each layer is used to collect, summarize, and/or transform data before passing them to the following layer. The DenseNet169 network [12], trained for the IMageNet Large Visual Recognition Challenge [13], and based on the analysis of non-radiological images, was used in this study. DenseNet architecture shows layers that are all directly connected with each other. This main advantage grants features reuse from different levels, thus improving computational, memory efficiency and focus on the problem of interest. DenseNet was computationally adapted to produce models for prosthesis failure recognition by using transfer learning and fine tuning algorithms [14]. Transfer learning consists in reusing a model developed for a task as the starting point for a model on a second task, i.e., some layers are frozen from the original task, while others are trained for the new one. It is a common and effective strategy to use transfer learning from a network pretrained on an extremely large dataset and then reuse it for a different task of interest, usually containing less images. The underlying assumption of transfer learning is that generic features learned on a large enough dataset can be shared among similar or even different datasets thus improving generalization.”

Figure 4(a). Authors mentioned that it is a healthy prosthesis and no regions where lighted up, but in the bottom-right side there is a red zone that far from the prosthesis. What happened here? other similar images presented the same results. Additionally, what is the explanation for the three light blue circles around the prosthesis in the same Figure 4(a)? I can appreciate a similar light blue circle in the top-right side of Figure 4(c).

R: GradCAM are used to highlight which parts of the image mostly contributed in the classification: in Figure 4(a), the non-failed prosthesis has no area peculiar for failure, thus it seems that the bottom-right corner was the most important area in the classification. In the color-code, red means the most important areas, whereas blue is the least important ones. Thus, light blue areas should be ignored.

The quality of Figures 2 o 3 must be improved.

R: We changed the figures.

Reviewer 2 Report

Thank you for allowing me to review the manuscript "AUTOMATIC IDENTIFICATION OF FAILURE IN HIP REPLACEMENT: AN ARTIFICIAL INTELLIGENCE APPROACH." The studies address a central topic of Deep Learning, namely the classification of images. Therefore, the findings largely coincide with studies on cancer diagnostics. 

Some small tips that should still be considered:

CNN training should be described in more detail. Especially the fine tuning and the algorithms. Furthermore, further statistical evaluation of specificity and sensitivity should be performed. 

In the discussion, the 5 misclassified cases should be further explained (characteristics).

Author Response

Thank you for allowing me to review the manuscript "AUTOMATIC IDENTIFICATION OF FAILURE IN HIP REPLACEMENT: AN ARTIFICIAL INTELLIGENCE APPROACH." The studies address a central topic of Deep Learning, namely the classification of images. Therefore, the findings largely coincide with studies on cancer diagnostics.

Some small tips that should still be considered:

CNN training should be described in more detail. Especially the fine tuning and the algorithms. Furthermore, further statistical evaluation of specificity and sensitivity should be performed.

R: We added information on CNN in the CNN training section, that now reads “A CNN is able to successfully capture the spatial dependencies in an image, through the application of relevant filters in parallel, and to classify based on them. A CNN is composed of various types of layers (or building blocks): each layer is used to collect, summarize, and/or transform data before passing them to the following layer. The DenseNet169 network [12], trained for the IMageNet Large Visual Recognition Challenge [13], and based on the analysis of non-radiological images, was used in this study. DenseNet architecture shows layers that are all directly connected with each other. This main advantage grants features reuse from different levels, thus improving computational, memory efficiency and focus on the problem of interest. DenseNet was computationally adapted to produce models for prosthesis failure recognition by using transfer learning and fine tuning algorithms [14]. Transfer learning consists in reusing a model developed for a task as the starting point for a model on a second task, i.e., some layers are frozen from the original task, while others are trained for the new one. It is a common and effective strategy to use transfer learning from a network pretrained on an extremely large dataset and then reuse it for a different task of interest, usually containing less images. The underlying assumption of transfer learning is that generic features learned on a large enough dataset can be shared among similar or even different datasets thus improving generalization.”

For the training and validation sets, we added accuracy and loss in Figures XX (a) and (b) respectively. These two metrics are important to be assessed as they give information on possible overfitting of the model.

In the discussion, the 5 misclassified cases should be further explained (characteristics).

R: We added information on these 5 cases, that now reads “Despite the good performance of the present algorithm, still 5 images were misclassified. In one case the algorithm did not detect the polyethylene insert wearing, indirectly represented in the x-ray film by an eccentric head in relation to the cup. A possible ex-planation for this is that polyethylene insert wearing as main reason for revision was an infrequent cause for failure in our data set (0,01%), therefore the algorithm may have been not trained enough to recognize this type of failure, and moreover in this specific case the degree of wearing was not as severe as the other cases present in the dataset. In another case the bone cerclage wiring around the prothesis was classified as pathological by the algorithm, instead the implant was not failed. In each of the remaining 3 cases the al-gorithm missed to identify a cup mobilization. A possible explanation is that in 2 of these cases the mistake was done on the lateral view, where the cup mobilization was harder to be identified.”

Reviewer 3 Report

1.In this paper, the authors have development an automated radiographic failure detection system. Methods: patients with 16 THA were included in the study, two third of them needed total or partial revision for prosthetic  loosening. The analysis is based on one antero-posterior and one lateral radiographic view obtained  from each patient during routine post-surgery follow-up.

 2. The paper is well-organized and easy to follow. There are a good introduction . Besides, the authors have presented a detailed and excellent discussion of the results. 

However, the author needs to address following minor comments:

1.Abstract needs revision should be included some number of the optimum results.

2.In the introduction section, the authors have not paid enough to the details of their work.

3.Why the first sentences of the abstract didn't give any feature about the work?

4.Generally, the article has attractive features, but the methodology is very limited. However, the authors can improve the readability of the conclusion and developed the clarity of results.

5.Fig.3 should be put names axes (y-axis) .

6.Line 179 what is meaning a, b, and a=1.

7.Why the authors didn't cite the enough new update references for the introduction part?

8.Why the authors didn't pay for their work sufficiently in the introduction section .

9.Mention the country origin of apparatuses.

10.Why the authors didn't pay more about the conclusion? the conclusion data are confusing.

Author Response

Reviewer 3

1.In this paper, the authors have development an automated radiographic failure detection system. Methods: patients with 16 THA were included in the study, two third of them needed total or partial revision for prosthetic  loosening. The analysis is based on one antero-posterior and one lateral radiographic view obtained  from each patient during routine post-surgery follow-up.

  1. The paper is well-organized and easy to follow. There are a good introduction . Besides, the authors have presented a detailed and excellent discussion of the results.

R: We thank the Reviewer for the appreciation of our work.

However, the author needs to address following minor comments:

1.Abstract needs revision should be included some number of the optimum results.

R: we added the sensitivity and specificity in the abstract, that now reads: “Results: After the training procedures, in the test set, the classification accuracy was 0.97, the sensitivity 0.97, the specificity 0.97 and the ROC AUC was 0.99”

2.In the introduction section, the authors have not paid enough to the details of their work.

R: We expanded the detail of the aim of our work and now reads: “The aim of this study was to develop the DL algorithm and the related preprocessing pipeline to automatically detect hip prosthetic loosening from conventional plain radio-graph and to employ the algorithm on a cohort of radiographs to analyze its performance in terms of sensitivity, specificity and accuracy.”

3.Why the first sentences of the abstract didn't give any feature about the work?

R: The first sentence was modified accordingly and now reads “This work is an attempt to use an artificial intelligence approach to solve an identified clinical need in the orthopedic field: the necessity of an early recognition of Total hip arthroplasty (THA) failure.”

4.Generally, the article has attractive features, but the methodology is very limited. However, the authors can improve the readability of the conclusion and developed the clarity of results.

R: concerning the methodology we expanded the section on the CNN training section. Concerning the discussion, we explained more deeply the 5 misclassified cases.

5.Fig.3 should be put names axes (y-axis) .

R: We added label for the y-axis.

6.Line 179 what is meaning a, b, and a=1.

R: We are sorry, they were a typing error, we removed all.

7.Why the authors didn't cite the enough new update references for the introduction part?

R: because besides being 10 years old we think they are still relevant to the topic and act as main basis for our work.

8.Why the authors didn't pay for their work sufficiently in the introduction section .

R: We expanded the detail of the aim of our work and now reads: “The aim of this study was to develop the DL algorithm and the related preprocessing pipeline to automatically detect hip prosthetic loosening from conventional plain radio-graph and to employ the algorithm on a cohort of radiographs to analyze its performance in terms of sensitivity, specificity and accuracy.”

9.Mention the country origin of apparatuses.

R: did the reviewer meant were the programs we used as Tensorflow 1.15 or Gradient weighted Class Activation Mapping (Grad-CAM) came from? Because no other apparatus was mentioned in the article

10.Why the authors didn't pay more about the conclusion? the conclusion data are confusing.

R: we changed the main sentence reporting the results. Now it reads: “The algorithm developed in this study showed a sensitivity of 96,7%, a specificity of 96,7% and an accuracy of 96,7%.” We added Figure 6 to better illustrate the failed cases of the algorithm.

Round 2

Reviewer 1 Report

The quality of the paper was improved, the authors provide now a more complete explanation about the working of the Neural Network algorithm, I accept it, despite of it is a little encyclopedic explanation without specific details of the implementation.

The absent of a cover letter for reviewers is not good. I had to compare the paper versions and figure out the changes.

I found two figures number 6 in the manuscript. This must be fixed.

Authors provided a sufficient explanation about the figure 6 and why the heatmap colored some regions in no failed prosthesis.

Reviewer 3 Report

-All comments made by authors ,only note:fig.6(The 5 misclassified images.) repeated must be become  fig7and it needs more details such as defined a,b,c,d,and e images.